# Carnitine insufficiency is associated with fatigue during lenvatinib treatment in patients with hepatocellular carcinoma

**Hironao Okubo**[1]*, **Hitoshi Ando**[2], **Kei Ishizuka**[1], **Ryuta Kitagawa**[1], **Shoki Okubo**[1], **Hiroaki Saito**[1], **Shigehiro Kokubu**[1,3], **Akihisa Miyazaki**[1], **Kenichi Ikejima**[4], **Shuichiro Shiina**[4], **Akihito Nagahara**[4]

**1** Department of Gastroenterology, Juntendo University Nerima Hospital, Tokyo, Japan, **2** Department of Cellular and Molecular Function Analysis, Kanazawa University Graduate School of Medical Sciences, Kanazawa, Ishikawa, Japan, **3** Department of Gastroenterology, Shin-yurigaoka General Hospital, Kawasaki, Kanagawa, Japan, **4** Department of Gastroenterology, Juntendo University School of Medicine, Tokyo, Japan

* drokubo@juntendo-nerima.jp

## Abstract

### Background

Fatigue is a common adverse event during lenvatinib treatment in patients with hepatocellular carcinoma. One mechanism contributing to development of fatigue might involve abnormal adenosine triphosphate synthesis that is caused by carnitine deficiency. To address this possibility, we examined the relationship between carnitine levels and fatigue during lenvatinib treatment.

### Methods

This prospective study evaluated 20 patients with hepatocellular carcinoma who underwent lenvatinib treatment. Both blood and urine samples were collected from the patients before starting lenvatinib therapy (day 0), and on days 3, 7, 14, and 28 thereafter. Plasma and urine concentrations of free and acyl carnitine (AC) were assessed at each time point. The changes in daily fatigue were evaluated using the Brief Fatigue Inventory (BFI).

### Results

Plasma levels of free carnitine (FC) at days 3 and 7 were significantly higher compared with baseline ($p = 0.005$, $p = 0.005$, respectively). The urine FC level at day 3 was significantly higher compared with baseline ($p = 0.030$) and that of day 7 tended to be higher compared with baseline ($p = 0.057$). The plasma AC concentration at days 14 and 28 was significantly higher compared with that of baseline ($p = 0.002$, $p = 0.005$, respectively). The plasma AC-to-FC (AC/FC) ratio on days 14 and 28 was significantly higher compared with baseline ($p = 0.001$, $p = 0.003$, respectively). There were significant correlations between the plasma AC/FC ratio and the change in the BFI score at days 14 and 28 ($r = 0.461$, $p = 0.041$; $r = 0.770$, $p = 0.002$, respectively).

**Data Availability Statement:** All relevant data are within the paper and its Supporting Information file.

**Funding:** The authors received no specific funding for this work.

**Competing interests:** The authors have declared that no competing interests exist.

## Conclusions

Longitudinal assessments of carnitine and fatigue in patients with hepatocellular carcinoma suggest that lenvatinib affects the carnitine system in patients undergoing lenvatinib therapy and that carnitine insufficiency increases fatigue. The occurrence of carnitine insufficiency may be a common cause of fatigue during the treatment.

## Introduction

Lenvatinib is a tyrosine kinase inhibitor (TKI) that was approved in Japan in 2018 for first-line treatment of unresectable hepatocellular carcinoma. Unexpected fatigue during lenvatinib treatment has been reported to be present in 30%–75% of the general hepatocellular carcinoma patient population [1, 2]. Fatigue is one of the most common causes of treatment discontinuation because of the patient's subsequent appetite loss, worsened quality of life, depression, and anxiety [1, 2]. The occurrence of fatigue during lenvatinib therapy has been reported to be higher compared with sorafenib, another TKI that is used to treat patients with hepatocellular carcinoma [3]. The alleviation of lenvatinib-induced fatigue is, therefore, an increasingly important clinical unmet need in the treatment of patients with hepatocellular carcinoma. However, factors that cause lenvatinib-induced fatigue have not been clarified.

Fatigue is a multidimensional and complex clinical symptom, which involves abnormal adenosine triphosphate synthesis [4]. Patients with cancer occasionally experience fatigue and general malaise before and after treatment, which is described as cancer-related fatigue. Although multifactorial and subjective cancer-associated conditions are influenced, decreased serum carnitine levels in patients who are undergoing cisplatin chemotherapy have been reported to be associated with fatigue [5–8]. Carnitine has an important biological function in the transport of long-chain fatty acids into the mitochondria for subsequent β-oxidation [9]. It also plays a role in eliminating acetyl carnitine, which is formed from excess accumulation of acetyl CoA because of decreased ATP synthesis [9]. The kidneys play an important role in maintaining carnitine homeostasis by reabsorbing appropriately 90% of the filtered carnitine at the level of the renal tubular epithelial cells [10,11]. Carnitine/organic cation transporter (OCTN)2, a sodium-dependent high affinity carnitine transporter, is strongly expressed in kidney, skeletal muscles, heart and placenta [12]. It is most abundant in the brush-border membrane of the kidney proximal tubules, where it is mainly involved in carnitine reabsorption [12,13]. Recent studies have demonstrated that modulation of the OCTN2 transport activity by administered OCTN2 substrate drugs such as oxaliplatin, verapamil, spironolactone, imatinib, and valproate can cause drug-induced secondary carnitine deficiency [13–15]. Additionally, a recent *in vitro* study showed that TKIs including imatinib, sorafenib, and sunitinib can inhibit OCTN2 function by appropriately 11%, 23%, and 54%, respectively, because of direct competitive inhibition of human OCTN2 [16].

The mechanism by which lenvatinib administration affects carnitine homeostasis remains unknown, and no studies have estimated the relationship between fatigue and carnitine levels during lenvatinib therapy. Therefore, we aimed to analyze plasma and urine carnitine levels and investigate the relationship between these levels and fatigue. Additionally, we explored the role of oral levocarnitine supplementation on fatigue during the lenvatinib treatment in a preliminary study.

## Materials and methods

### Patients and treatments

This was a single-center, exploratory, prospective study. This study was approved by the Ethical Review Board of Juntendo University Faculty of Medicine (Jundai-Irin No-2016135) and was performed in accordance with the 1964 Declaration of Helsinki and its later amendments. Written informed consent was obtained from all patients before enrollment. The study period was from February 2019 through November 2019. Patients who were going to receive lenvatinib therapy for unresectable hepatocellular carcinoma were included. No patients who had a history of previous molecular-targeted agent treatment were included. Patients with impaired Eastern Cooperative Oncology Group performance status (PS 2, 3, and 4) were also excluded from the study. Patients received oral lenvatinib (Eisai Co., Ltd., Tokyo, Japan) 8 mg/day (for bodyweight <60 kg) or 12 mg/day (for bodyweight ≥60 kg) once daily after breakfast.

Adverse event (AE) grades were assessed using the Common Terminology Criteria for Adverse Events (CTCAE) version 4.0. In accordance with the guidelines for lenvatinib administration, the drug dose was reduced when a patient developed any grade 3 or 4 AE, or any unacceptable CTCAE 4.0 grade 2 AE. The Brief Fatigue Inventory (BFI), a nine-item questionnaire for the assessment of fatigue severity, was used to evaluate fatigue [17,18]. Patients with fatigue of over CTCAE 4.0 Grade 2, which is equivalent to a BFI score of over 3–5 using the questionnaire, after 2 or 4 weeks of lenvatinib therapy were administrated levocarnitine 1500 mg per day orally. Those who did not report worse fatigue compared with the baseline received no supplementation. Thyroid function was tested before initiation of lenvatinib therapy and monitored bi-weekly thereafter, and thyroid-replacement medication was adjusted as necessary.

Fasting morning blood and urine samples were collected from patients before starting lenvatinib treatment (day 0), and on the days 3, 7, 14, and 28. Total carnitine and free carnitine (FC) levels in the plasma and urine in all specimens were measured by an enzyme cycling method using an autoanalyzer (JCA-BM8040 series; JEOL, Tokyo, Japan) and acyl carnitine (AC) was calculated as the difference between total carnitine and FC [19]. For the analysis, the urinary carnitine concentration was divided by the urinary creatinine concentration to correct for the influence of dilution. The plasma AC-to-FC (AC/FC) ratio was calculated based on the measured values. Therapeutic response was evaluated using enhanced computed tomography that was obtained 6 weeks after starting lenvatinib, in accordance with the modified Response Evaluation Criteria in Solid Tumors (RECIST) [20].

### Statistical analysis

The Wilcoxon signed-rank test was used to compare between the plasma and urinary baseline carnitine levels and each time point (days 3, 7, 14, and 28). To identify differences between the baseline plasma AC/FC ratio and those of each time point after oral lenvatinib administration, a paired sample *t*-test was used after the data were tested for normal distribution using the Shapiro–Wilk normality test. Pearson's correlation coefficient was used to determine the association between BFI and the AC/FC ratio. All statistical analyses were performed using SPSS Statistics for Windows, Version 22 (IBM Corp., Tokyo, Japan). All tests were two-sided, and $p$ values < 0.05 were considered to be statistically significant.

## Results

### Patient characteristics and dynamic changes in carnitine concentration

Twenty Japanese patients (five females and 15 males) with a median age of 76 years (range, 63–88 years) were enrolled into the study. The clinical profiles of all patents included in this

study are shown in Table 1. The etiology of hepatocellular carcinoma was hepatitis B virus antigen-positive in two patients, hepatitis C virus antibody-positive in eight patients and other causes in 10 patients. The objective response was complete response (CR) in one patients, partial response (PR) in 11 patients, stable disease (SD) in five patients and progressive disease (PD) in three patients, respectively, with an overall response rate of 60% and a disease control rate of 85%.

The dynamic changes in plasma and urine FC and AC levels after the start of lenvatinib treatment are illustrated in Fig 1. The data after initiation of oral levocarnitine supplementation and cessation of the lenvatinib after day 14 were excluded. The plasma FC concentration on days 3 and 7 was significantly higher compared with the baseline carnitine concentrations ($p = 0.005$, $p = 0.005$, respectively) and it returned to near-normal levels 14 and 28 days after the start of therapy (Fig 1A). However, the plasma AC concentration on days 14 and 28 were significantly higher compared with baseline ($p = 0.002$, $p = 0.005$, respectively). The urine FC-to-creatinine ratio on day 3 was significantly higher compared with baseline levels ($p = 0.030$) and that on day 7 tended to be higher than that of baseline, but the difference was not significant ($p = 0.057$). However, it subsequently returned to normal 14 and 28 days after the start of therapy (Fig 1B). Additionally, as illustrated in Fig 2, we found that the percent increase from day 0 in urine to day 7 was significantly correlated with that in plasma ($r = 0.545$, $p = 0.013$).

The individual time course of the plasma AC/FC ratio in 20 patients treated with lenvatinib, excluding the data after levocarnitine supplementation or cessation of the lenvatinib, is shown in Fig 3. The mean and standard error of the plasma AC/FC ratio on days 0, 3, 7, 14, and 28 were 0.207 ± 0.061, 0.165 ± 0.055, 0.185 ± 0.079, 0.292 ± 0.107, and 0.279 ± 0.092, respectively. The value on day 3 was significantly lower compared with baseline ($p = 0.024$). However, the ratio on days 14 and 28 was significantly higher compared with baseline ($p = 0.001$, $p = 0.003$,

**Table 1. Patient characteristics.**

| Characteristic | N = 20 |
|---|---|
| Age, median (years) | 76 (63–88) |
| Male/Female | 15/5 |
| ECOG 0/1 | 19/1 |
| Body weight, median, (kg) | 59.0 (38.6–77) |
| Etiology HBV/HCV/NBNC | 2/8/10 |
| Starting dose 12/8 mg | 8/12 |
| Child-Pugh score 5/6 | 12/8 |
| BCLC staging B/C | 20/5 |
| Extra hepatic spread Yes/No | 4/16 |
| ALBI grade 1/2a/2b | 4/10/6 |
| Albumin, median (g/dL) | 3.7 (3–4.2) |
| Total bilirubin, median (mg/dL) | 0.8 (0.4–1.6) |
| Estimated glomerular filtration rate (mL/min/1.73 m2) | 61.7 (43.2–101.5) |
| Prothrombin time, median (%) | 88 (68–105) |
| NH3, median (μg/dL) | 34 (16–112) |
| AFP, median (ng/mL) | 26.6 (0.9–57200) |
| PIVKA-II, median (MAU/mL) | 1470 (15–68600) |
| Objective response CR/PR/SD/PD | 1/11/5/3 |

ECOG: Eastern Cooperative Oncology Group; HBV: hepatitis B virus; HCV; hepatitis C virus; NBNC: non-hepatitis B or C virus; ALBI: Albumin-Bilirubin; AFP: alpha-fetoprotein; PIVKA-II: protein induced by vitamin K absence or antagonist; CR: complete response; PR: partial response, SD: stable disease; PD: progressive disease.

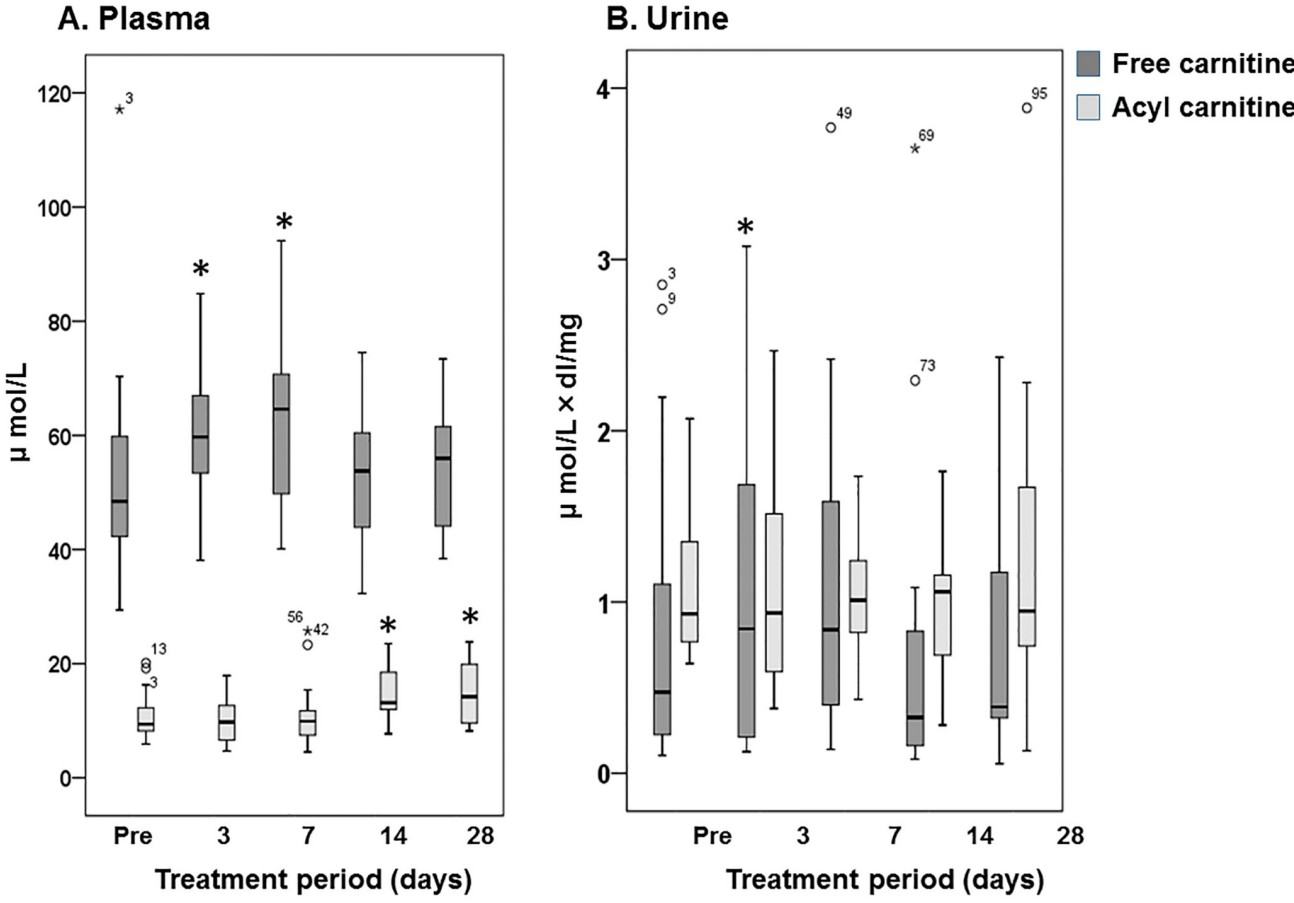

**Fig 1. Changes in free and acyl carnitine in plasma creatinine (A) and urine, as assessed by urinary creatinine (B) after starting lenvatinib treatment.** The box spans data between two quartiles (IQR, interquartile range), and the bold line represents the median. The ends of the whiskers represent the largest and smallest values that are not outliers. Outliers are values between 1.5 and 3 IQRs from the end of the box. * $p < 0.05$; versus baseline (Pre).

respectively). On day 28, overt carnitine insufficiency based on the AC/FC ratio >0.4 [21, 22] was found in 3/20 (15%) patients.

## Relationship between the plasma AC/FC ratio and the Brief Fatigue Inventory

We then assessed the correlation between the plasma AC/FC ratio and BFI. At baseline, there was a nearly-significant positive correlation between the two variables (Fig 4A; r = 0.410, p = 0.073). We also found that there was a significant positive correlation between the AC/FC ratio and the BFI differences for each patient between day 14 and day 28 (Fig 4B and 4C; r = 0.461, p = 0.041; r = 0.770, p = 0.002, respectively).

## Changes in the Brief Fatigue Inventory and plasma AC/FC ratio after oral levocarnitine supplementation

During the lenvatinib on or after 4 weeks of the administration, 11 patients (55%) reported fatigue of over CTCAE Grade 2, which was equivalent to a BFI score of greater than 3–5, and the patients received oral levocarnitine supplementation (1500 mg per day) after reporting the maximum fatigue. The change in the BFI and plasma AC/FC ratio among baseline, the day

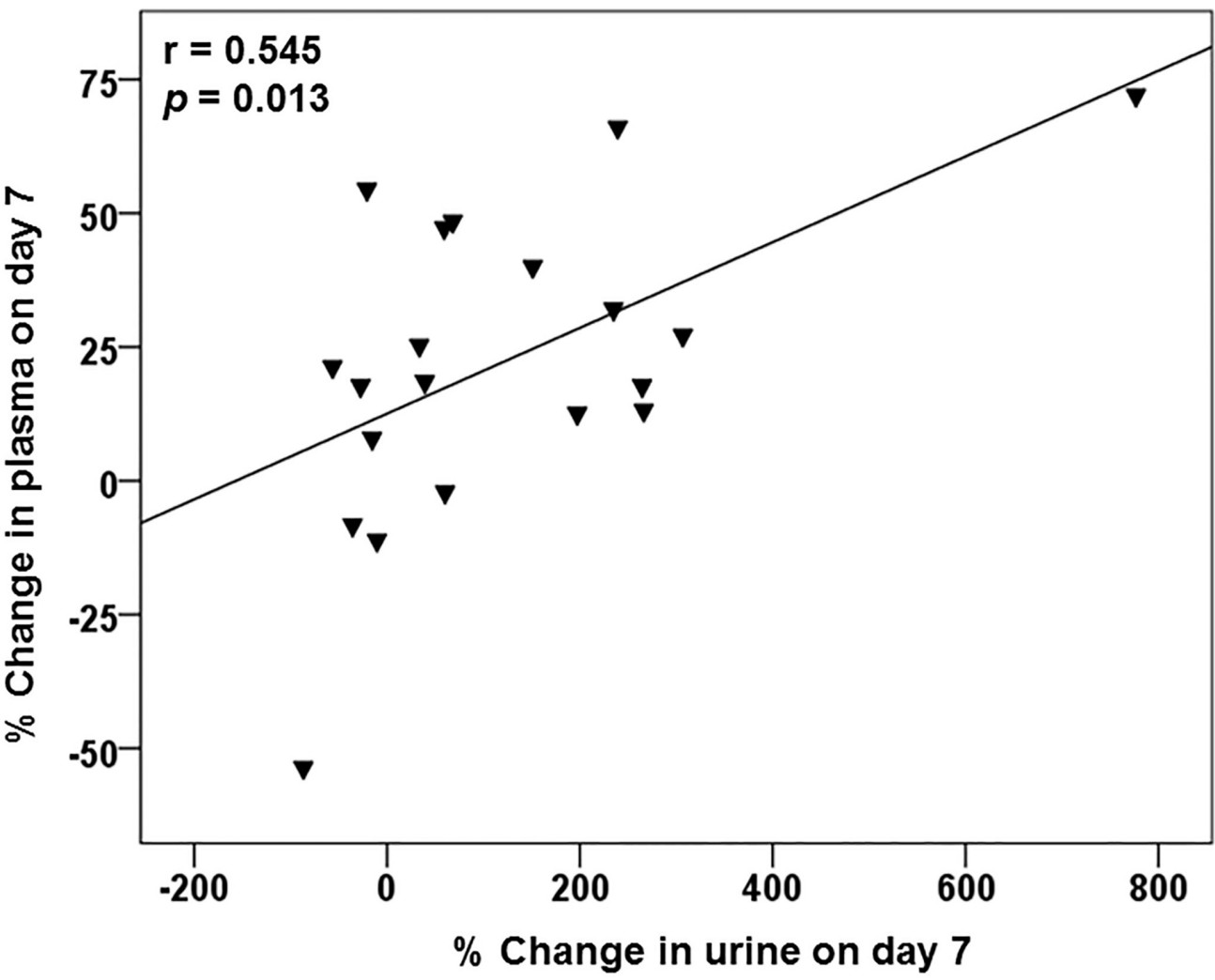

**Fig 2. Correlation between the change in plasma and urine free carnitine on day 7 after starting lenvatinib therapy.** r = 0.545, $p$ = 0.013.

before levocarnitine supplementation, and 4 weeks after the supplementation is shown in Fig 5. The BFI on the day before levocarnitine supplementation, which was significantly increased compared with baseline ($p < 0.001$), was significantly reduced at 4 weeks after starting the carnitine supplementation ($p < 0.001$). Seven of 11 patients (64%) had a dose reduction, but the BFI score improved in four of 11 patients (36%) without a lenvatinib dose reduction. Although the plasma AC/FC ratio on the day before levocarnitine supplementation significantly increased compared with baseline ($p = 0.002$), there was no significant difference between that of the day before supplementation and that of 4 weeks after supplementation ($p = 0.984$).

## Discussion

In this study, we have shown that lenvatinib has the potential to affect plasma and urinary carnitine levels after the start of oral administration.

First, we found that plasma and urinary FC levels increased for 3–7 days after starting lenvatinib administration and the increase in urinary carnitine excretion was related to that of the

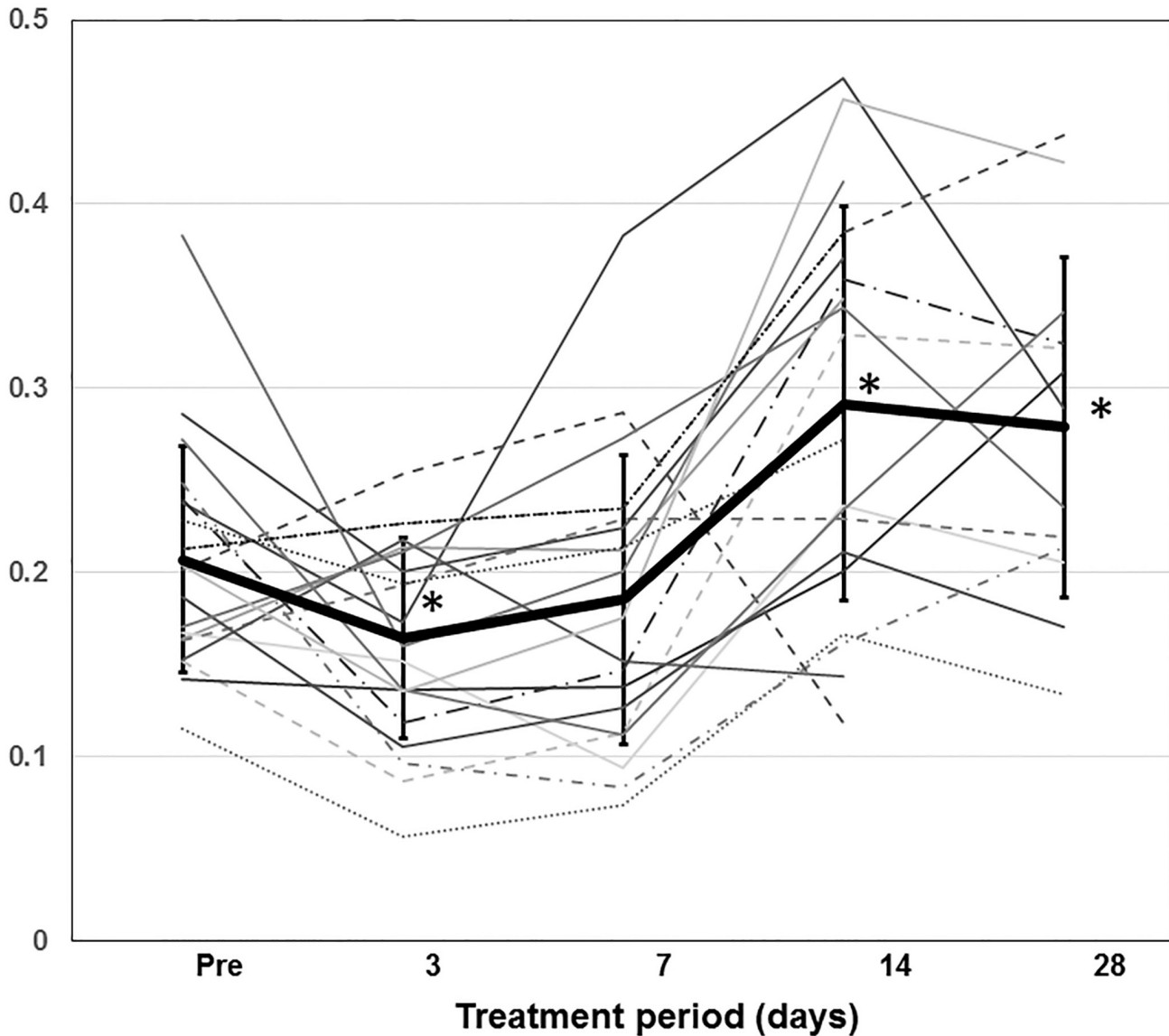

**Fig 3. Time profiles of the plasma acyl-to-free carnitine ratio after starting lenvatinib treatment.** The heavy line represents the mean with the standard error. AC/FC ratio, AC-to-FC ratio. * $p < 0.05$; versus baseline (Pre).

plasma carnitine levels. Our results suggest that disposition of plasma carnitine is the same as that of urinary carnitine at an early stage after the start of lenvatinib therapy. Second, we showed that, despite no remarkable changes in urinary carnitine levels at steady state such as at days 14 and 28, the plasma AC/FC ratio, which serves as an indicator of carnitine deficiency, increased because of the increase in the plasma AC concentration. Although the mechanism for fluctuation of carnitine disposition is unclear, cellular or intra-mitochondrial uptake of carnitine might be inhibited by lenvatinib administration in terms of increase in plasma carnitine and the increased plasma carnitine levels would lead to increased urinary excretion. Given the carnitine shuttle [23], there are two major possible explanations for the increase in plasma AC

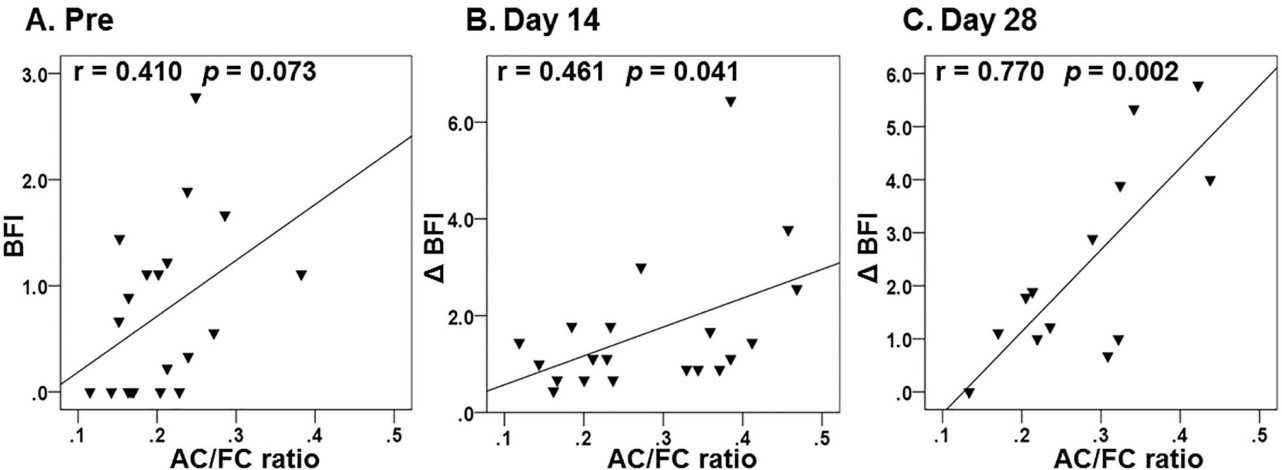

**Fig 4. Correlation between the plasma acyl-to-free carnitine ratio and the Brief Fatigue Inventory (BFI) for each patient between baseline (A) and day 14 (B) or 28 (C).** AC/FC ratio, AC-to-FC ratio; BFI, Brief Fatigue Inventory; ΔBFI, difference in the Brief Fatigue Inventory.

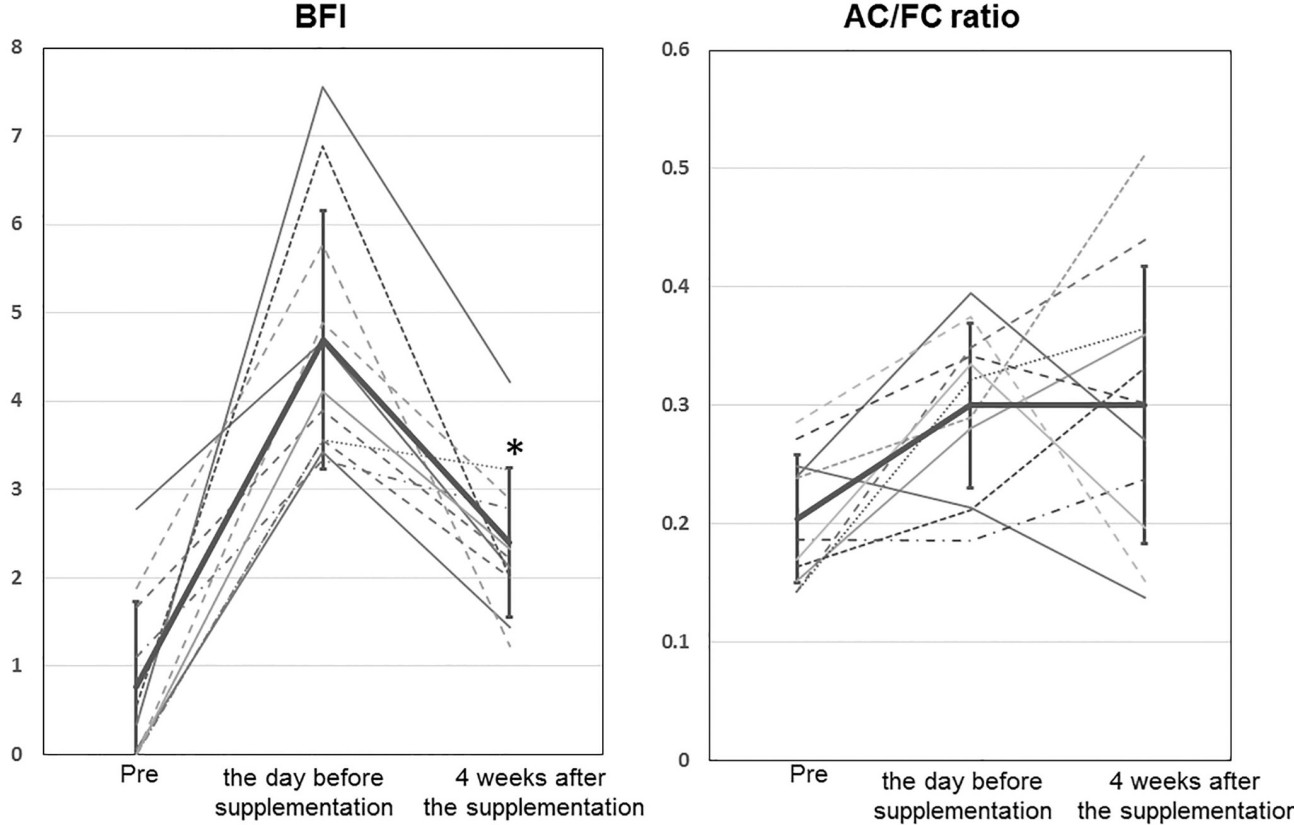

**Fig 5. Change in the Brief Fatigue Inventory and the plasma acyl-to-free carnitine (AC/FC) ratio among baseline, the day before levocarnitine supplementation, and 4 weeks after the supplementation.** The thick line represents the mean with the standard error. The solid line represents without a lenvatinib dose reduction, and the dotted line represents with a lenvatinib dose reduction. BFI, Brief Fatigue Inventory; AC/FC ratio, AC-to-FC ratio; * $p < 0.05$ versus the day before levocarnitine supplementation.

at steady state. One explanation is that intra-mitochondrial uptake of AC may be reduced if long-chain AC is increased. The second explanation is that acetyl CoA may accumulate in the skeletal muscles and acetyl carnitine metabolism may be impaired in hepatocytes if acetyl carnitine is increased.

Additionally, in this study, we showed that the plasma AC/FC ratio is related to the severity of fatigue. There was a strong correlation between BFI and AC/FC ratio at day 28 existed regardless of the presence or absence of viral hepatitis, whether it was a viral infection or other causes (r = 0.840 and r = 0.773, respectively).

Under normal conditions, 80% of the total plasma carnitine is FC and 20% is AC, with a normal AC/FC ratio of 0.25 [24, 25]. A plasma AC/FC >0.4 is considered to be abnormal and represents carnitine insufficiency, as previously reported, and this is also included in the Japanese guidelines [21, 22]. Our results show that 15% of the patients were in the category at day 14 and most patients may have latent carnitine insufficiency 14 days after the start of lenvatinib. Anorexia caused by anticancer therapy may reduce oral intake, which may lead to low plasma carnitine levels [5]. Because all the included patients did not report appetite loss until 28 days after starting lenvatinib, an increased AC/FC ratio would mainly reflect the development of fatigue.

Previous studies have identified several widely used anticancer drugs as OCTN2 inhibitors, including vinblastine, actinomycin D, and etoposide [15, 16]. Hu et al. reported, in an *in vitro* study, that sorafenib, the first-line TKI for treating patients with hepatocellular carcinoma, can inhibit the OCTN2 function by approximately 23% because of direct competitive inhibition of human OCTN2 [16]. Because lenvatinib is also a kinase inhibitor, it may inhibit the OCTN's function, but *in vitro* studies on lenvatinib inhibition of OCTN function have not been performed. However, because more fatigue was reported during lenvatinib therapy compared with sorafenib [3], it is possible that lenvatinib's inhibitory effect on OCTN2 is stronger compared with that of sorafenib.

It is crucial for clinicians to manage lenvatinib-induced fatigue, continue the therapy, and achieve a long treatment duration that may lead to improvement in the patient's survival period by maintaining the relative dose intensity [26]. The finding that lenvatinib may be more or less associated with secondary drug-induced carnitine insufficiency would lead to an improvement in the quality of pharmacological treatment. To the best of our knowledge, this is the first study to demonstrate the association between carnitine dynamics and fatigue during lenvatinib therapy in patients with hepatocellular carcinoma.

It is plausible that restoration of plasma carnitine levels via carnitine supplementation could ameliorate lenvatinib-induced fatigue. Recent studies showed a decrease in fatigue with levocarnitine supplementation in patients with various types of cancer [7, 27]. In this study, levocarnitine administration in hepatocellular carcinoma patients who are undergoing lenvatinib therapy improved the BFI score, as expected. It should be noted that the BFI score improved in four of 11 patients (36%) without a lenvatinib dose reduction and the AC/FC ratio was decreased in three of four patients.

An improvement in the patients' fatigue would be affected by both carnitine supplementation and various other factors including the placebo effect and tumor shrinkage that result from the antitumor effect of lenvatinib. The AC/FC ratio was not necessarily improved in some patients after the supplementation. This may be because the placebo effect influences the BFI score during lenvatinib treatment. Based on a recent meta-analysis, there is not enough evidence to support the recommendation to use carnitine supplementation for cancer-related fatigue in the oncological clinical setting [28]. However, a substantial improvement in fatigue that was observed in patients without a lenvatinib dose reduction during levocarnitine supplementation provides strong support for our finding that lenvatinib therapy caused carnitine

insufficiency. The efficacy of carnitine supplementation to patients who were administered lenvatinib requires confirmation in a randomized controlled trial. In addition, it is important to consider the type of patients who are recommended to receive carnitine supplementation. Because the baseline plasma AC/FC ratio of non-responders (SD and PD) tended to be higher compared with that of responders (CR and PR) ($p = 0.057$), it might be better to consider levo-carnitine supplementation in patients with a high AC/FC ratio to retain the relative lenvatinib dose intensity.

Throughout the study period, only the plasma AC/FC ratio among all of the markers (plasma FC, plasma AC, urine FC, urine AC levels, and urine AC/FC ratio) was associated with BFI (data not shown). Therefore, only the plasma AC/FC ratio seems to be a useful marker during lenvatinib therapy. As mentioned above, the baseline plasma AC/FC ratio seems to be useful for predicting incomplete treatment with lenvatinib.

Our study has several limitations. First, the number of patients included is limited. Second, because carnitine analyses were measured using an enzyme cycling method, it is not possible to determine whether the increase in AC was long-chain AC or acetyl carnitine. Third, only fatigue was analyzed in this study. Adrenal function, which is reportedly a cause of fatigue during the treatment, was not examined [29]. Despite these limitations, our findings may provide evidence that lenvatinib affects carnitine homeostasis, and they suggest that carnitine insufficiency is a major factor in the development of fatigue during the therapy.

In conclusion, our findings suggest that lenvatinib therapy is associated with drug-induced carnitine insufficiency. Therefore, monitoring blood carnitine levels during treatment might be useful for anticipating the development of fatigue. Further research is needed to clarify the role in levocarnitine supplementation for preventing or improving fatigue.

## Supporting information

**S1 Data. Patient characteristics and data.**
(XLSX)

## Acknowledgments

The authors thank K. Nakamura, N. Akita, and M. Morinaga for their excellent technical assistance. We thank Jodi Smith, PhD, from Edanz Group (www.edanzediting.com/ac) for editing a draft of this manuscript.

## Author Contributions

**Conceptualization:** Hironao Okubo, Hitoshi Ando.

**Data curation:** Hironao Okubo, Kei Ishizuka, Ryuta Kitagawa, Shoki Okubo, Hiroaki Saito.

**Formal analysis:** Hironao Okubo, Hitoshi Ando.

**Investigation:** Hironao Okubo, Kei Ishizuka, Ryuta Kitagawa, Shoki Okubo, Hiroaki Saito.

**Methodology:** Hironao Okubo, Hitoshi Ando, Shigehiro Kokubu.

**Project administration:** Hironao Okubo, Hitoshi Ando.

**Supervision:** Akihisa Miyazaki, Kenichi Ikejima, Shuichiro Shiina, Akihito Nagahara.

**Writing – original draft:** Hironao Okubo, Hitoshi Ando.

**Writing – review & editing:** Hironao Okubo, Hitoshi Ando, Akihisa Miyazaki, Kenichi Ikejima.

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
