## [Decision Letter · Decision Letter 0]

10 Dec 2019

PONE-D-19-32377

Carnitine insufficiency is associated with fatigue during lenvatinib treatment in patients with hepatocellular carcinoma

PLOS ONE

Dear Dr. Okubo,

Thank you for submitting your manuscript to PLOS ONE. After careful consideration, we feel that it has merit but does not fully meet PLOS ONE’s publication criteria as it currently stands. Therefore, we invite you to submit a revised version of the manuscript that addresses the points raised during the review process.

This manuscript seems to be important in this area.

1. Authors should describe the status of HBsAg and anti-HCV in table 1, and discuss their influence.

2. Major problem is that the number of patients is too small, and authors should increase the number of patients to at least 20. 

We would appreciate receiving your revised manuscript by Jan 24 2020 11:59PM. To enhance the reproducibility of your results, we recommend that if applicable you deposit your laboratory protocols in protocols.io, where a protocol can be assigned its own identifier (DOI) such that it can be cited independently in the future. For instructions see: http://journals.plos.org/plosone/s/submission-guidelines#loc-laboratory-protocols

We look forward to receiving your revised manuscript.

Kind regards,

Tatsuo Kanda, M.D., Ph.D.

Academic Editor

PLOS ONE

Journal Requirements:

Please ensure that your manuscript meets PLOS ONE's style requirements, including those for file naming. The PLOS ONE style templates can be found at http://www.plosone.org/attachments/PLOSOne_formatting_sample_main_body.pdf and http://www.plosone.org/attachments/PLOSOne_formatting_sample_title_authors_affiliations.pdf

2. Thank you for including your ethics statement in the manuscript Methods section:

'The study was approved by the ethics board of our university (Jundaiirin 2018193) and was performed in accordance with the 1964 Declaration of Helsinki and its later amendments.'

3. We noticed you have some minor occurrence(s) of overlapping text with the following previous publication(s), which needs to be addressed:

https://doi.org/10.1007/s00520-018-4521-6

https://doi.org/10.1371/journal.pone.0196747

https://doi.org/10.3109/00498254.2010.494201

In your revision ensure you cite all your sources (including your own works), and quote or rephrase any duplicated text outside the Methods section. Further consideration is dependent on these concerns being addressed.

Reviewers' comments:

Reviewer's Responses to Questions

**Comments to the Author**

1. Is the manuscript technically sound, and do the data support the conclusions?

Reviewer #1: Yes

Reviewer #2: Partly

2. Has the statistical analysis been performed appropriately and rigorously? 

Reviewer #1: Yes

Reviewer #2: Yes

3. Have the authors made all data underlying the findings in their manuscript fully available?

Reviewer #1: Yes

Reviewer #2: Yes

4. Is the manuscript presented in an intelligible fashion and written in standard English?

Reviewer #1: Yes

Reviewer #2: Yes

5. Review Comments to the Author

Reviewer #1: The authors described about carnitine defficiency due to administration of lenvatinib, and efficatiy of levocarnitine supplementation in patients with hepatocellular carcinoma. Although a long treatment duration of lenvatinib is important for improvement of the patients prognosis, it is not easy to manage various adverse effects. This study showed one of the way of overcoming lenvatinib related fatigue. However some problems remain before publish in this manuscript.

In Figure 4, the authors showed the correlation between BFI and AC/FC ratio in Day 14, and Day 28. Please add the figure of the correlation between BFI and AC/FC ratio at baseline. If correlation was not showed at baseline, please discuss the cause of this phenomenon.

Please insert reference to discussion section, line 256.

Reviewer #2: The authors tried to evaluate the relationship between carnitine levels and fatigue during lenvatinib treatment and found there were significant correlations between the plasma AC/FC ratio and the change in the BFI score at days 14 and 28. The paper is well written and organized, however, there are several concerns in the study.

Major comments.

As the authors mentioned in the limitations, the study cohort is too small. The reviewer would suggest that the authors evaluate ATP or energy charge level and/or OCTN functions of the patients. Detailed analysis using more parameters or translational experiments might add strength in the study regardless of the cohort size.

The authors should provide information about treatment response and its relationship with the carnitine levels.

The comparison of AC/FC levels between sorafenib and lenvatinib would be of interest. Although the BFI improved after the supplementation, AC/FC did not decrease. The authors conclude the well correlation between AC/FC and BFI during the treatment and they should discuss why this happened (why the AC/FC did not improve after the supplementation).

6. PLOS authors have the option to publish the peer review history of their article (what does this mean?). If published, this will include your full peer review and any attached files.

Reviewer #1: No

Reviewer #2: No

---

## [Author Response · Author response to Decision Letter 0]

14 Jan 2020

Response to the Editor

11 January 2020

Dr. Tatsuo Kanda

Academic Editor

PLOS ONE

Please find attached a copy of our revised manuscript entitled “Carnitine insufficiency is associated with fatigue during lenvatinib treatment in patients with hepatocellular carcinoma”, which we would like to resubmit for publication in PLOS ONE. 

We would like to thank the academic editor and the reviewers for their helpful comments. We have implemented the suggested changes and trust that our manuscript is now suitable for publication in your journal. Please find below a detailed description of the revisions made to the manuscript in our point-by-point responses to the comments from the reviewers.

Yours sincerely,

Hironao Okubo, MD, PhD 

Department of Gastroenterology 

Juntendo University Nerima Hospital 

Response to academic editor

1) Authors should describe the status of HBsAg and anti-HCV in table 1, and discuss their influence.

Response: Thank you for your valuable comments. In accordance with the comment, we have added information about the etiology, which includes the presence or absence of HBs-Ag/HCV-Ab, in Table 1 and the Discussion section (lines 256–258 as follows:

There was a strong correlation between BFI and AC/FC ratio at day 28 existed regardless of the presence or absence of viral hepatitis, whether it was a viral infection or other causes (r = 0.840 and r = 0.773, respectively).

2) Major problem is that the number of patients is too small, and authors should increase the number of patients to at least 20.

Response: In accordance with the advice, we added the two most recent cases, and the data from 20 patients were reanalyzed. The revised results further support our conclusion.

#Response to reviewer 1

1) In Figure 4, the authors showed the correlation between BFI and AC/FC ratio in Day 14, and Day 28. Please add the figure of the correlation between BFI and AC/FC ratio at baseline. If correlation was not showed at baseline, please discuss the cause of this phenomenon.

Response: In accordance with the comment, we added the data for the correlation between BFI and the AC/FC ratio at baseline to Figure 4 of the revised manuscript. We found that there was a nearly significant positive correlation between the two variables (Fig 4A; r = 0.410, p = 0.073). This description was added to the revised manuscript (lines 200-202).

2) Please insert reference to discussion section, line 256.

Response: In accordance with the reviewer’s advice, we have added the following reference to the Discussion (line 281 in the revised manuscript): Takahashi A, Moriguchi M, Seko Y, Ishikawa H, Yo T, Kimura H, et al. Impact of relative dose intensity of early-phase lenvatinib treatment on therapeutic response in hepatocellular carcinoma. Anticancer Res. 2019;39:5149-56. 

#Response to reviewer 2 

1. As the authors mentioned in the limitations, the study cohort is too small. The reviewer would suggest that the authors evaluate ATP or energy charge level and/or OCTN functions of the patients. Detailed analysis using more parameters or translational experiments might add strength in the study regardless of the cohort size.

Response: Thank you for the insightful and constructive comment. We agree with the comment, but evaluation of ATP/energy charge level or OCTN functions is extremely difficult in patients with cancer in clinical practice. Therefore, in accordance　with the academic editor’s advice, we added the additional two cases and re-analyzed the data for all 20 patients. We believe that our present clinical data are valuable and worth publishing in PLOS ONE.

2. The authors should provide information about treatment response and its relationship with the carnitine levels.

Response: In accordance with the comment, we have conducted an additional analysis about the relationship between the treatment response and carnitine levels. 

The treatment responses were added to Table 1, and we discussed the relationship in the revised manuscript (lines 305–310) as follows: In addition, it is important to consider the type of patients who are recommended to receive carnitine supplementation. Because the baseline plasma AC/FC ratio in non-responders (SD and PD) tended to be higher compared with that of responders (CR and PR) (p = 0.057), it might be better to consider levocarnitine supplementation in patients with a high AC/FC ratio to retain the relative lenvatinib dose intensity. 

3. The comparison of AC/FC levels between sorafenib and lenvatinib would be of interest. 

Response: We thank the Reviewer for this insightful comment. In accordance with this comment, we added the presumptive difference in the inhibitory effect on OCTN2 between lenvatinib and sorafenib, which would affect the AC/FC level, into the Discussion section of the revised manuscript (lines 275–278) as follows: However, because more fatigue was reported during lenvatinib therapy compared with sorafenib [3], it is possible that lenvatinib’s inhibitory effect on OCTN2 is stronger compared with that of sorafenib.

4. Although the BFI improved after the supplementation, AC/FC did not decrease. The authors conclude the well correlation between AC/FC and BFI during the treatment and they should discuss why this happened (why the AC/FC did not improve after the supplementation). 

Response: We would like to express our strong appreciation to reviewer 2 for this comment. As the reviewer pointed out, the AC/FC ratio did not necessarily improve after the levocarnitine supplementation. We think that the placebo effect, including the recognition of dose reduction, might influence the BFI score especially in some patients with dose reduction. We provided this interpretation on lines 296–298 of the revised manuscript. Moreover, for reader’s convenience, we have changed Figure 5, which separately shows patients with and without dose reduction. 

#Response to Journal Requirement

1. We have checked that our manuscript meets STROBE statement.

2. a) As pointed out, the notation has been changed as follows: This study was approved by the Ethics Review Board of Juntendo University Faculty of Medicine (Jundai-Irin No-2016135) and was performed in accordance with the 1964 Declaration of Helsinki and its later amendments.

b) In accordance with the comment, we have inserted the same text to the “Ethics Statement”.

3. Thank you for pointing some minor occurrences of overlapping text. The notation has been changed as follows. 

Lines 68–69: Patients with cancer occasionally experience fatigue and general malaise before and after treatment, which is described as cancer-related fatigue.

Line 98–99: This study was approved by the Ethics Review Board of Juntendo University Faculty of Medicine

Line 185–189: The box spans data between two quartiles (IQR, interquartile range), and the bold line represents the median. The ends of the whiskers represent the largest and smallest values that are not outliers.

---

## [Decision Letter · Decision Letter 1]

21 Jan 2020

PONE-D-19-32377R1

Carnitine insufficiency is associated with fatigue during lenvatinib treatment in patients with hepatocellular carcinoma

PLOS ONE

Dear Prof. Okubo,

Thank you for submitting your manuscript to PLOS ONE. After careful consideration, we feel that it has merit but does not fully meet PLOS ONE’s publication criteria as it currently stands. Therefore, we invite you to submit a revised version of the manuscript that addresses the points raised during the review process.

We would appreciate receiving your revised manuscript by Mar 06 2020 11:59PM. To enhance the reproducibility of your results, we recommend that if applicable you deposit your laboratory protocols in protocols.io, where a protocol can be assigned its own identifier (DOI) such that it can be cited independently in the future. For instructions see: http://journals.plos.org/plosone/s/submission-guidelines#loc-laboratory-protocols

We look forward to receiving your revised manuscript.

Kind regards,

Tatsuo Kanda, M.D., Ph.D.

Academic Editor

PLOS ONE

Reviewers' comments:

Reviewer's Responses to Questions

**Comments to the Author**

1. If the authors have adequately addressed your comments raised in a previous round of review and you feel that this manuscript is now acceptable for publication, you may indicate that here to bypass the “Comments to the Author” section, enter your conflict of interest statement in the “Confidential to Editor” section, and submit your "Accept" recommendation.

Reviewer #1: All comments have been addressed

Reviewer #2: All comments have been addressed

2. Is the manuscript technically sound, and do the data support the conclusions?

Reviewer #1: Yes

Reviewer #2: Yes

3. Has the statistical analysis been performed appropriately and rigorously? 

Reviewer #1: Yes

Reviewer #2: Yes

4. Have the authors made all data underlying the findings in their manuscript fully available?

Reviewer #1: Yes

Reviewer #2: Yes

5. Is the manuscript presented in an intelligible fashion and written in standard English?

Reviewer #1: Yes

Reviewer #2: Yes

6. Review Comments to the Author

Reviewer #1: Thank you for revising the problem.

The correlation coefficient between BFI and AC/FC ratio is increases from baseline to day 28 in Figure 4.

I think it is consistent because fatigue is relatively weak at baseline.

Revised manuscript satisfies my requirement.

Thank you.

Reviewer #2: The authors have fulfilled each of the major compulsory revisions and modified the manuscript as requested. I have the following further suggestions that in my opinion will improve the quality of the manuscript.

In the study, authors measured urine/plasma AC/FC. The reviewer was wondering if they could describe which marker was the most useful in the real-world clinical setting: the reason, speculation, and how often it should be monitored. Please add this information in the discussion section.

7. PLOS authors have the option to publish the peer review history of their article (what does this mean?). If published, this will include your full peer review and any attached files.

Reviewer #1: No

Reviewer #2: No

---

## [Author Response · Author response to Decision Letter 1]

3 Feb 2020

#Response to reviewer 2 

1. In the study, authors measured urine/plasma AC/FC. The reviewer was wondering if they could describe which marker was the most useful in the real-world clinical setting: the reason, speculation, and how often it should be monitored. Please add this information in the discussion section.

Response: Thank you for the insightful comment. In accordance with the comment, we discussed the utility of the plasma AC/FC ratio in the real-world clinical setting in the revised manuscript (lines 311–316) as follows: Throughout the study period, only the plasma AC/FC ratio among all of the markers (plasma FC, plasma AC, urine FC, urine AC levels, and urine AC/FC ratio) was associated with BFI (data not shown). Therefore, only the plasma AC/FC ratio seems to be a useful marker during lenvatinib therapy. As mentioned above, the baseline plasma AC/FC ratio seems to be useful for predicting incomplete treatment with lenvatinib.

---

## [Editor Report · Decision Letter 2]

5 Feb 2020

PONE-D-19-32377R2

Carnitine insufficiency is associated with fatigue during lenvatinib treatment in patients with hepatocellular carcinoma

PLOS ONE

Dear Prof. Okubo,

Thank you for submitting your manuscript to PLOS ONE. After careful consideration, we feel that it has merit but does not fully meet PLOS ONE’s publication criteria as it currently stands. Therefore, we invite you to submit a revised version of the manuscript that addresses the points raised during the review process.

Authors should make a list of each patients'characteristics and data by an excel file and submit the excel file as a supplementary file with manuscript again.

We would appreciate receiving your revised manuscript by Mar 21 2020 11:59PM. To enhance the reproducibility of your results, we recommend that if applicable you deposit your laboratory protocols in protocols.io, where a protocol can be assigned its own identifier (DOI) such that it can be cited independently in the future. For instructions see: http://journals.plos.org/plosone/s/submission-guidelines#loc-laboratory-protocols

We look forward to receiving your revised manuscript.

Kind regards,

Tatsuo Kanda, M.D., Ph.D.

Academic Editor

PLOS ONE

---

## [Author Response · Author response to Decision Letter 2]

12 Feb 2020

Response to the Editor

12 February 2020

Dr. Tatsuo Kanda, Academic Editor

PLOS ONE

Dear Dr. Kanda,

Please find attached a copy of our revised manuscript titled “Carnitine insufficiency is associated with fatigue during lenvatinib treatment in patients with hepatocellular carcinoma”, which we would like to resubmit for publication in PLOS ONE. 

We would like to thank for the helpful comment. We have added a supplementary file including the patient characteristics and data, and we trust that our manuscript is now suitable for publication in your journal. 

Yours sincerely,

Hironao Okubo, MD, PhD 

Department of Gastroenterology 

Juntendo University Nerima Hospital 

#Response to the academic editor 

1. Authors should make a list of each patients' characteristics and data by an excel file and submit the excel file as a supplementary file with manuscript again.

Response: Thank you for the helpful comment. In accordance with the comment, we have added a supplementary file including the patient characteristics and data, and we trust that our manuscript is now suitable for publication in your journal. In addition, we have added the supporting information into the revised manuscript (lines 330–331) as follows: 

Supporting information

S1 Data. Patient characteristics and data.

---

## [Editor Report · Decision Letter 3]

14 Feb 2020

Carnitine insufficiency is associated with fatigue during lenvatinib treatment in patients with hepatocellular carcinoma

PONE-D-19-32377R3

Dear Dr. Okubo,

We are pleased to inform you that your manuscript has been judged scientifically suitable for publication and will be formally accepted for publication once it complies with all outstanding technical requirements.

With kind regards,

Tatsuo Kanda, M.D., Ph.D.

Academic Editor

PLOS ONE
---

## [Editor Report · Acceptance letter]

19 Feb 2020

PONE-D-19-32377R3 

Carnitine insufficiency is associated with fatigue during lenvatinib treatment in patients with hepatocellular carcinoma 

Dear Dr. Okubo:

I am pleased to inform you that your manuscript has been deemed suitable for publication in PLOS ONE. Congratulations! Your manuscript is now with our production department. 

With kind regards,

on behalf of

Dr. Tatsuo Kanda 

Academic Editor

PLOS ONE